# The Role of CD1 Gene Polymorphism in the Genetic Susceptibility to Spondyloarthropathies in the Moroccan Population and the Possible Cross-Link with Celiac Disease

**DOI:** 10.3390/vaccines11020237

**Published:** 2023-01-20

**Authors:** Angelica Canossi, Khadija Oumhani, Tiziana Del Beato, Pierluigi Sebastiani, Alessia Colanardi, Anna Aureli

**Affiliations:** 1National Research Council (CNR), Institute of Translational Pharmacology (IFT), 67100 L’Aquila, Italy; 2Laboratoire d’Immunologie, Institut National d’Hygiene, Rabat 10000, Morocco

**Keywords:** spondyloarthropathy, HLA gene polymorphisms, CD1 genes, IBDs

## Abstract

Spondyloarthropathies (SpA) are a group of chronic inflammatory disorders usually affecting the axial spine and asymmetrical peripheral joints. Strong evidence links genetic and environmental factors to SpA pathogenesis. The HLA-B27 is the most important genetic factor associated with SpA. Nevertheless, the involvement of other HLA and non-HLA loci has been also reported. Some patients with SpA may also manifest features of celiac disease (CeD), thus suggesting a genetic overlap across these autoimmune diseases. Recently, CD1 glycoproteins, a class of molecules able to bind and present non peptidic antigens to T cells, aroused interest for their contribution to the pathogenesis of CeD. Therefore, to evaluate whether functional polymorphisms of CD1A and E genes also influence susceptibility to SpA, we analyzed 86 patients from Morocco affected by SpA and 51 healthy controls, using direct sequencing analysis. An increase of CD1E*01/01 homozygous genotype (*p* = 0.046) was found in SpA, compared with controls. CD1E*01/01 genotype was associated particularly to patients with sacroiliac joints/spine/peripheral joints pain (*p* = 0.0068), while a decrease of CD1E*01/02 genotype was evidenced compared to controls (*p* = 0.0065). Results from haplotypes analysis demonstrated that CD1A*02-E*02 decreased the risk of SpA, while CD1A*02-E*01 increased risk to develop disease. Our data indicate a relationship between CD1 genes and susceptibility to SpA in the Moroccan population and suggest the existence of shared genetic risk loci across SpA and CeD that might be useful to explain common pathogenetic features and define novel therapeutic strategies.

## 1. Introduction

Spondyloarthropathies (SpA) are a group of heterogeneous immune-mediated inflammatory diseases, comprising ankylosing spondylitis (AS), axial spondyloarthritis (AxSpa), non-radiographic axial spondyloarthritis (nr-axSpA), peripheral spondyloarthritis (pSpA), reactive arthritis (ReA) and enteropathic arthritis (EnA) [1,2].

An inflammation at the attachment sites of ligaments, tendons and joint capsules to the bone has been detectedat the origin of SpA [3] The axial skeleton, more specifically the spine and sacroiliac joint, is the main site affected [3].

Uveitis, psoriasis and inflammatory bowel diseases (IBDs) represent the most typical extra-articular manifestations of SpA [3,4]. Occurrence of IBDs or other gastrointestinal diseases such as celiac disease (CeD) has been associated with EnA [5,6], thus suggesting the existence of a possible relationship between inflammation of the gut mucosa and arthritis [7,8,9].

Prevalence of SpA in the Western population is estimated approximately 0.5–1.5% [3] while in North Africa [10,11] and the Middle East region [11,12] is less common. However, few data are available from these countries where low human leukocyte antigen (HLA)-B27 frequency [13] is considered the principal reason to low prevalence of SpA [14]. Indeed it is well-known that HLA-B27 in the major histocompatibility complex (MHC) locusis the major genetic risk factor linked to SpA [15,16]. Nevertheless, besides MHC, additional genetic factors of susceptibility to SpA have been proposed [17], whose influence remains to be confirmed.

CD1 glycoproteins aroused our interest for their role in presenting lipid antigens to T cells. CD1 is a small multigene family consisting of five MHC-like genes (CD1A, -B, -C, -D, and -E) located on the q22 arm of chromosome 1. CD1 proteins present self and foreign lipid antigens to specific T and natural killer T (NKT) cells [18], which are important in controlling autoimmune diseases, tumor growth, and host defense against pathogens [19]. Interestingly, it has been proposed that the interaction CD1-restricted NKT cells during early life microbial exposures could have an impact on later susceptibility to immune/inflammatory diseases [20]. Scientific evidence shows a link between these genes and inflammatory neuropathies [21] and multiple sclerosis (MS) [22].

As described in Porcelli et al., these molecules are heterodimers composed of a heavy chain non covalently bound to β2-microglobulin and which comprises three protein domains (α1, α2 and α3) [23]. Two of these, α1 and α2 domains, form the antigen-binding groove that is deeper and narrower than that of MHC class I molecules and is structured in pockets containing non-polar amino acids which facilitate the binding of the hydrophobic portion of the glycolipids. Instead, α3 domain is associated to the β2-microglobulin [23,24].

The limited polymorphism is the hallmark of CD1 molecules with respect to MHC class I and II molecules; CD1E represents the only exception with six alleles described so far [25,26].

In a previous work, we have already studied these molecules to establish their involvement in CeD risk in a Moroccan population. Our data suggested a positive association between CD1 genes, particularly of CD1E, and CeD predisposition in that population [27].

Therefore, here we considered investigating polymorphisms in exon 2 of CD1A and –E in a group of patients with SpA from the same geographic area, to verify the hypothesis of a possible genetic overlap between SpA and CeD.

## 2. Materials and Methods

In total, eighty-six Moroccan patients with SpA (65 men and 21 women) were received from different rheumatology departments of the University Hospital Center (CHU) of Rabat—Salé according to the modified international criteria of New York; they were included in the study and analyzed for CD1polymorphisms.

Diagnosis of SpA was confirmed by clinical manifestations; patients with uncertain diagnosis were excluded from the study.

The genetic study was approved by the scientific committee of the National Center for Scientific and Technical research of Morocco (Code 103212) with respect to the ethical principles of the Ministry of health of Morocco and informed consent was obtained from all individuals.

Fifty-one healthy Moroccan individuals with no history of SpA or other autoimmune diseases were used for comparison.

### 2.1. Analysis of CD1A, and CD1E Gene Polymorphism

Genomic DNA was obtained from peripheral blood (PB) cells by a column-based nucleic acid purification method (QIAGEN DNA Blood Mini kit, Qiagen, Hilden, Germany).

Exon 2 of CD1A and E genes was amplified by polymerase chain reaction (PCR) using primers previously described [27]. PCR amplifications consisted of 50ng of genomic DNA, 1X PCR buffer with MgC_l2_, 0.2 mM each of deoxynucleotides, 0.5 U of Taq polymerase and a 3.2 pmol/μL concentration of each primer in a final volume of 50 μL. PCR cycling conditions were: 5 min denaturation at 95 °C, 30 cycles of 20 s at 95 °C, 30 s at 60 °C, 30 s at 72 °C and a final elongation cycle of 5 min at 72 °C. PCR products were identified on 1.5% agarose gel electrophoresis and subsequently purified by a PCR clean-up reagent (EXOSAP).

Sequence reactions were performed using the Big Dye Terminator Chemistry v 1.1 (Applied Biosystems, Foster City, CA, USA) and processed on an ABI Prism 3130 Genetic Analyzer (Applied Biosystems) and then purified. Typing was obtained based on alignments of the processed sequences with exon2 sequences of the human CD1A and -E genes retrieved from the Genbank.

### 2.2. Statistical Analysis

Allele and genotype frequencies were obtained by direct counting. CD1 haplotype frequencies were estimated using an expectation–maximization method for multilocus genotypic data when the gametic phase is not known [28]. Results of allele/genotype/haplotype frequencies were analyzed using Pearson’s chi-square with Yate’s correction or Fisher’s Exact test with Bonferroni’s correction (pc), as appropriate, and compared with those of controls. The odds ratio (OR) and 95% confidence interval (CI) were computed. The level of significance was set at *p* = 0.05. SPSS statistics software andArlequin v3.5 population genetics software(http://cmpg.unibe.ch/software/arlequin3, 10 October 2022) [29] were used for analyses.

## 3. Results

To determine whether some polymorphisms of CD1 loci are associated with increased or reduced susceptibility to develop SpA, the CD1A and E genotype and allele frequencies of 86 Moroccan SpA patients and 51 controls from the same geographic area were evaluated.

The 86 cases included in the study were stratified in 3 subgroups based on reported symptoms; subgroup1—sacroiliac joints/spine/peripheral joints pain (n = 64; 74.4%); subgroup 2—pleuropulmonary abnormalities (*n* = 10; 11.6%); subgroup 3—extra-articular manifestations (*n* = 12; 14%) (Figure 1).

CD1A genotype/allele frequencies have not shown differences among patients and controls (Table 1).

A different CD1E *genotype distribution* was instead observed between SpA patients and the healthy subjects (Table 1). CD1E*01/01 homozygous was significantly more frequent in patients with SpA than in controls (Table 1:42% vs. 23%, *p* = 0.046, OR:2.28, CI:1.0770 to 5.0843). Particularly, as shown in Table 2, CD1E*01/01 frequency was significantly increased in the subgroup 1 compared with controls (50% vs. 23%, *p* = 0.0068, OR:3.25,CI: 1.4434 to 7.3176). This result was validated by Bonferroni’s correction for multiple comparison and *p* value remained significant (*p* = 0.034). Moreover, individuals carryingCD1E*01/02 genotype were significantly decreased among patients of subgroup 1, compared with healthy subjects (28% vs. 55%, *p* = 0.0065, OR:0.32, CI:0.1480 to 0.6979). OR indicates that the presence of this genotype drops the risk for SpA and *p* value remained significant (pc = 0.0325) also after Bonferroni’s correction.

No statistically significant differences in CD1 E genotype/allele frequencies have been detected between the other two subgroups of patients and healthy individuals.

Furthermore, a significantly greater combination of CD1A* 02/02 and CD1E* 01/01 genotypes was observed in patients in subgroup 1 compared to controls (46% vs. 23%, *p* = 0.02, OR:2.77, CI:1.2271 to 6.2621).

Allele frequencies were in Hardy–Weinberg equilibrium at each locus both for SpA patients (CD1A, *p* = 1.000; CD1E, *p* = 0.228) and controls (CD1A, *p* = 1.000; CD1E, *p* = 0.700).

Analysis of estimated CD1A-CD1E haplotypes demonstrated a remarkably lower prevalence of CD1A*02-E*02 in the group of patients with sacroiliac joints/spine/peripheral joints pain than in controls (30% vs. 43%; the two-tailed *p* value equals 0.053). Furthermore, a higher prevalence of CD1A*02-E*01 (65% vs. 52%, *p* = 0.057) was evidenced in this SpA subtype than in controls.

Interestingly to note that in subgroup 3 of SpA patients, those with extra-articular manifestations, CD1A*02-E*02 had an opposite trend (58%) compared to the other subgroups of patients (vs. 30% of subgroup 1 and 25% of subgroup 2 (*p* = 0.01 and *p* = 0.05, respectively) (Table 3).

## 4. Discussion

Growing evidence indicates that gut microbiome is involved in the pathogenesis of SpA [7] and AS patients may have a 5.3-fold higher incidence of chronic inflammatory intestinal diseases such as IBD than healthy individuals [30]. IBD has a partially shared pathogenesis with CeD; in both conditions, genetics and environment contribute to immune dysregulation, leading to chronic inflammation and disease [31].

HLA region is strongly involved both in SpA [16] and in CeD susceptibility [32], while it shows only a weak/moderate association with IBD [33].However, the presence of HLA-B27 or DQ2 or DQ8 genes is not a “conditio sine qua non” for SpA and CeD development respectively; indeed, not all the individuals who have such genetic risk factorsdevelop these diseases. Therefore, it is believed that additional host genetic factors control these diseases susceptibility and pathogenesis, and it could be hypotesized that some of these may be shared among SpA and CeD.

Some research showed a relationship between CD1 polymorphisms and some autoimmune and inflammatory diseases. Particularly, our previous study indicated the existence of an association between CD1 and the risk of CeD in a population from Morocco [27].

Based on these observations, here we investigated the contribution of CD1 to SpA in a cohort of patients and healthy individuals from the same geographic area.

Our research highlights a different distribution of CD1E genotypes between SpA patients and control subjects. The CD1E*01/01 homozygous genotype was significantly associated to SpA risk, particularly inpatients with pain localized to the axial spine and sacroiliac and peripheral joints. Conversely, CD1E*01/02 resulted associated to a *lower risk* of SpA development.

Furthermore, a detailed study of CD1A and E genotype combinations allowed to identify that the combination of CD1A*02/02 and CD1E*01/01 genotypes has an additional predisposing effect to develop SpA. Interestingly, the combination of CD1A*02/02 and CD1E*02/02 genotypes was significantly increased among patients with extra-articular manifestations.

It is unknown whether CD1A*02/02 and CD1E*02/02 associated genotypes affect risk of SpA and their impact in the presence of SpA with extra-articular manifestations remains relatively unexplored. However, given that our previous study demonstrated that this genotype combination was also increased in patients with CeD from Morocco, we could speculate that the presence of CD1A* 02/02 and CD1E* 02/02 associated genotypes supports the existence of the link between these two diseases.

Studies on human group 3 CD1antigens, which includes CD1e [34,35,36] demonstrated that, unlike other CD1 proteins, CD1e can promote the presentation of glycolipid antigens by CD1b [35]. By operating a lipid and glycolipids antigen selection to be presented by other CD1 isoforms, CD1e might modulate both antimicrobial immune response and autoimmunity.

Chronic inflammatory diseases have been recently linked to changes in gut microbiome composition and function and CeD could be considered an example [37]. In CeD, CD1d could present Bacteroides-derived sphingolipids to NKT cells, thus regulating immune response. In recent times, the influence of the binding between NKT cells and CD1-restricted lipids in dysregulating the immune system-microbiota crosstalk and in the development of chronic inflammatory diseases has been studied [38]

The prototype ligand of iNKT cells is the α-galactosylceramide (α-GalCer), a sphingolipid that, as shown in a mouse experimental model, can activate NKT cells to suppress myelin antigen-specific Th1 responses, thus protecting susceptible mice against experimental autoimmune encephalomyelitis (EAE), a valuable model for MS [39].

However, it has been also reported that high doses of α-GalCer, directly interact with CD1d expressed on T cells and potentiate EAE by facilitating the Th17 and Th1 differentiation and survival [39].

In addition, in mouse models it has also been shown that gut microbiota directs Th17 cell differentiation in the lamina propria of the small intestine [40]. Then, intestinal Th17 cells, induced by gut-residing segmented filamentous bacteria, produce IL-17 which is considered as a mediator in SpA pathogenesis [41,42].

From this point of view, it could be supposed that some CD1E gene variants, modifying binding affinity of lipid antigens, might continue to be selected for their potential role in protective immunity against microbial glycolipids.

To the best of our knowledge, this is the first demonstration that CD1 genes are associated to SpA predisposition. With our previous study on a population from Morocco, we have already suggested the importance to consider CD1 genes as possible markers in CeA risk and evaluate their involvement also in malaria protection. The data presented herein demonstrate that, in the same geographic area, CD1 polymorphisms can be linked also to the SpA development. In particular, we would speculate that CD1A* 02/02 and CD1E* 01/01 genotype combination could be a common predisposing factor both for SpA and CeD.

Although our findings have some limitations, mainly attributable to the risk of bias due to the constitution of non-homogeneous patient groups and of limited size, we hypothesize that our more stringent eligibility criteria may help to study populations with less risk of errors. Moreover, the results obtained from our study are supported by multiple comparison tests. In fact, in spite of the Bonferroni’s correction, the positive associations remained significant.

## 5. Conclusions

In conclusion, our work suggests that CD1 genes could be involved in the predisposition of SpA in the Moroccan population. However, we believe that it may be important to deeply investigate the role of these genes as translational markers both across autoimmune and infective diseases and improve knowledge on CD1-targeted T-cell therapies.

Keeping in mind that autoimmune diseases, including SpA, arise from a combination of genetic predisposition and environmental factors and that the genetic risk is determined by the interplay of hundreds of genes [43,44,45,46,47,48,49,50] (Figure 2), we would like to highlight the importance of carrying out more studies in other ethnic groups and with larger sample sizes. Advances in genetic analysis will permit both to confirm a pathogenic link between CD1 genes and SpA and also to find other genetic variants that make significantly contributions to risk disease. The identification of novel therapeutic targets could aid the development of new treatments and allow SpA patients to benefit from them.

## Figures and Tables

**Figure 1 vaccines-11-00237-f001:**
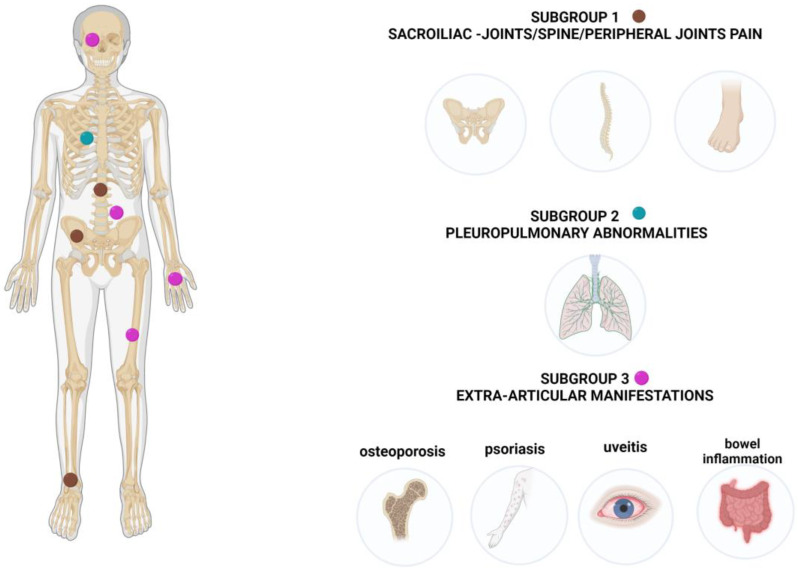
Subgroups of patients based on clinical presentation.

**Figure 2 vaccines-11-00237-f002:**
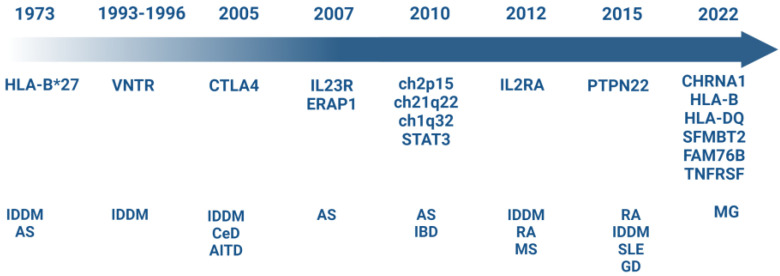
Timeline of principal risk genetic loci associated to the development of autoimmune diseases from Seventies to present [43,44,45,46,47,48,49,50]. An updated list of studies and associations can be found in the genome-wide association studies (GWAS) catalog (https://www.ebi.ac.uk/gwas/).

**Table 1 vaccines-11-00237-t001:** CD1A and CD1E genotypes and alleles in patients with SpA and healthy individuals from Morocco.

	All SpA Patients		Controls		*p* Value	OR
CD1A genotypes	*n* = 84 *	**%**	*n* = 51	**%**		
CD1A*01/01	0	0.0	0	0.0	ns	-
CD1A*01/02	8	9.5	3	6.0	ns	-
CD1A*02/02	76	90.5	48	94.0	ns	-
Alleles	2*n* = 168	%	2*n* = 102	**%**		
CD1A*01	8	4.8	3	2.9	ns	-
CD1A*02	160	95.2	99	97.1	ns	-
CD1EGenotypes	*n* = 86	%	*n* = 51	**%**		
CD1E*01/01	36	41.9	12	23.5	*p* = 0.046	2.34 ^a^
CD1E*01/02	33	38.3	28	54.9	ns	-
CD1E*02/02	16	18.6	9	17.6	ns	-
CD1E*01/05 CD1E*02/05	1 0	1.2 0.0	1 1	2.0 2.0	ns ns	-
Alleles	2*n* = 172	%	2*n* = 102	**%**		
CD1E*01	106	61.6	53	52.0	ns	-
CD1E*02	65	37.8	47	46.0	ns	-
CD1E*05	1	0.6	2	2.0	ns	-

* Two samples were not available; ^a^ 95% CI: 1.0770 to 5.0843.

**Table 2 vaccines-11-00237-t002:** CD1A and CD1E genotypes and alleles in patients of subgroup 1 and healthy individuals from Morocco.

	SpA1 Patients		Controls		*p* Value	OR
CD1Agenotypes	*n* = 62 *	**%**	*n* = 51	**%**		
CD1A*01/01	0	0.0	0	0.0	ns	-
CD1A*01/02	6	9.7	3	5.9	ns	-
CD1A*02/02	56	90.3	48	94.1	ns	-
Alleles	2*n* = 124	**%**	2*n* = 102	**%**		
CD1A*01	6	4.8	3	2.9	ns	-
CD1A*02	118	95.2	99	97.1	ns	-
CD1Egenotypes	*n* = 64	**%**	*n* = 51	**%**		
CD1E*01/01	32	**50.0**	**12**	**23.5**	***p*** = 0.0068	3.25 ^a^
CD1E*01/02	18	28.1	28	54.9	***p*** = 0.0065	0.32 ^b^
CD1E*02/02	13	20.3	9	17.6	ns	-
CD1E*01/05 CD1E*02/05	1 0	1.6 0.0	1 1	2.0 2.0	ns ns	- -
Alleles	2*n* = 128	**%**	2*n* = 102	**%**		
CD1E*01	83	64.8	53	52.0	ns	-
CD1E*02	44	34.4	47	46.0	ns	-
CD1E*05	1	0.8	2	2.0	ns	-

* Two samples were not available; ^a^ 95% CI: 1.4434 to 7.3176; ^b^ 95% CI: 0.1480 to 0.6979.

**Table 3 vaccines-11-00237-t003:** CD1A-CD1E estimated haplotypes in patients with SpA and in control populations.

	SpA Total (2*n* = 168)	SpA1 ^a^ (2*n* = 124)	SpA2 ^b^ (2*n* = 20)	SpA3 ^c^ (2*n* = 24)	HC (2*n* = 102)
*CD1A-CD1Ehaplotypes*	*hf* ^d^	*hf*	*hf*	*hf*	*hf*
CD1A*02-CD1E*01	0.62	0.65 ^§^	0.65	0.41	0.52 ^§^
CD1A*02-CD1E*02	0.33	0.30 °	0.25	0.58	0.43 °
CD1A*01-CD1E*02	0.04	0.04	0.10	-	0.03
CD1A*01-CD1E*05	0.01	0.01	-	-	-
CD1A*02-CD1E*05	-	-	-	-	0.02
CD1A*01-CD1E*01	-	-	-	-	-

^a^ SpA1 patients with sacroiliac joints/spine/peripheral joints pain; ^b^ SpA2 patients with pleuropulmonary abnormalities; ^c^ SpA3 patients with extra-articular manifestations; ^d^ Haplotype frequencies; ° *p* = 0.053; ^§^
*p* = 0.057.

## Data Availability

All data, code, and materials used in the analysis are available to any researcher for purposes of reproducing or extending the analysis andare available in the main text.

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
