# Peer review of "The Role of CD1 Gene Polymorphism in the Genetic Susceptibility to Spondyloarthropathies in the Moroccan Population and the Possible Cross-Link with Celiac Disease"

_vaccines, 2023, doi:10.3390/vaccines11020237_

Round 1
Reviewer 1 Report
The research try to elucidate the possible role the of CD1 gene polymorphism in the genetic susceptibility to spondyloarthropaties and the possible link with celiac disease in the Moroccan population;
The research is relevant in the field trying to bridge a link between spondyloarthropaties and celiac diseases;
With a view to understanding the associations between different genetic diseases, the work adds new epidemiologic knowledge of CD1 gene polymorphism in the Moroccan population;
The research seems to be conducted with rigorous methodology;
The conclusion are robust and validate the hypothesis of association between the polymorphisms of the examined genes and the possible correlation with studied phenotypes;
The reference are appropriate in terms of number and linked with the proposed study;
The figures appear to be explanatory and the tables adequate.
Author Response
We thank the Reviewer for having appreciated our manuscript and for the positive comments.
Reviewer 2 Report
This manuscript, written by Dr. Canossi, original research, with the title of "The role of CD1 gene polymorphism in the genetic susceptibility to spondyloarthropaties in the Moroccan population and the possible cross-link with celiac disease", analyzed the CD1 gene in a series of 86 patients.
The term spondyloarthritis (SpA) is used for a family of disorders, including ankylosing spondylitis (AS, radiographic axial spondyloarthritis [r-axSpA]), nonradiographic axial spondyloarthritis (nr-axSpA), forms of arthritis associated with psoriasis and with inflammatory bowel diseases, and other conditions. Compared with the general population, patients with SpA have higher frequencies of the human leukocyte antigen (HLA) B27, and of sacroiliitis by radiography or magnetic resonance imaging (MRI).
The manuscript is well written, it is easy to read and understand.
Comments:
1) In the Tables. There is a column with % data. Could you please confirm that the numbers shown there are percentages? For example, CD1A genotypes (n = 84) are 0.0, 9.5, and 90.5. But in CD1E genotypes (n = 86) are 0.42, 0.38, and 0.19.
2) It is stated that multiple comparisons and Bonberroni correction was made. As I understand about the Bonberroni correction, for example in Table 1 there are 13 comparisons. For Bonberroni it would be 0.05 exp13 ? Based on the type of data and number of cases, I think that if you perform multiple comparision correction (Bonberroni or false discovery rate), all p values will be negative. In this type of analysis, no correction would me more adequate.
3) Could you please explain who the data of table 3 was calculated?
4) Could you please add a table or figure showing the different markers associated with the autoimmune diseases? Studies of WGAS, review types, show this data.
Author Response
We thank the Reviewer for the positive comments. According to her/his suggestions, our revisions are indicated below;
- In the Tables. There is a column with % data. Could you please confirm that the numbers shown there are percentages? For example, CD1A genotypes (n = 84) are 0.0, 9.5, and 90.5. But in CD1E genotypes (n = 86) are 0.42, 0.38, and 0.19.
We provided to check all columns with % data and make corrections (Lines 160-165).
- It is stated that multiple comparisons and Bonberroni correction was made. As I understand about the Bonberroni correction, for example in Table 1 there are 13 comparisons. For Bonberroni it would be 0.05 exp13 ? Based on the type of data and number of cases, I think that if you perform multiple comparision correction (Bonberroni or false discovery rate), all p values will be negative. In this type of analysis, no correction would me more adequate.
Bonferroni's correction has been applied to significative P values of CD1E. On the basis of the CD1E genotypes, five comparisons have been considered.
- Could you please explain who the data of table 3 was calculated?
Data of table 3 have been calculated by Dr. Canossi and Dr. Aureli using Arlequin v3.5 population genetics software.
4) Could you please add a table or figure showing the different markers associated with the autoimmune diseases? Studies of WGAS, review types, show this data.
According to the Reviewer’s suggestion, we added in the new “Conclusion” section, a figure (figure 2) in which principal genetic risk loci associated to several autoimmune diseases have been depicted along a time's arrow from seventies to present.